# Hydrochemical Characteristics and Controlling Factors of Surface Water in Upper Nujiang River, Qinghai-Tibet Plateau

**Mingguo Wang** [1,2,3], **Li Yang** [2,*], **Jingjie Li** [2] **and Qian Liang** [3]

1  College of Chemistry, Zhengzhou University, Zhengzhou 450001, China; wangmingguo@mail.cgs.gov.cn
2  Center for Hydrogeology and Environmental Geology Survey, CGS, Baoding 071051, China; lijingjie@mail.cgs.gov.cn
3  Henan Nonferrous Metals Geological Exploration Institute, Zhengzhou 450052, China; qianl996@163.com
*  Correspondence: yli_b@mail.cgs.gov.cn

**Abstract:** Rivers play an essential role in the global matter transport and hydrogeochemical cycle. Hydrogeochemical research is significant to developing and protecting water resources in the Qinghai-Tibet Plateau and its lower reaches. This study aimed to identify the hydrochemical characteristics and controlling factors of Daqu River and Gaqu River in Dingqing County, two tributaries of the upper Nujiang River. This study used spatial analysis, trilinear diagram analysis, and ion ratio analysis of hydrochemical indexes to investigate the hydrochemical characteristics and controlling factors. Results show that $Ca^{2+}$ and $Mg^{2+}$, and $HCO_3^-$ and $SO_4^{2-}$ are the main cations and anions of these two rivers. $HCO_3 \cdot SO_4$-$Mg \cdot Ca$ and $HCO_3 \cdot SO_4$-$Ca \cdot Mg$ are the primary hydrochemical types for the surface water of Daqu and Gaqu Rivers, respectively. The influence of atmospheric precipitation and anthropogenic activities is weak. The carbonated water–rock reaction is the main $Ca^{2+}$, $Mg^{2+}$, and $HCO_3^-$ source, and hot springs act as the primary source of $SO_4^{2-}$ and supplements $Ca^{2+}$, $Mg^{2+}$, and $HCO_3^-$. $Mg^{2+}$ from magnesite dissolution exists in the Daqu River basin. Weak reverse cation exchange occurs in both rivers. Daqu River receives more low-mineralized glacier meltwater along the flow, whereas Gaqu River receives more high-mineralized hot spring water.

**Keywords:** hydrochemistry; ion source; ion ratio; water–rock interaction; upper Nujiang River



## 1. Introduction

Rivers are one of the most important and active components of the geochemical system, which is important to both the physical geographical system and the human–Earth relationship [1]. As an important channel for the transport of matter and energy, rivers are an essential part of the global hydrogeochemical cycle and play an important role in regulating ecosystem balance and climate change [2–4]. By analyzing the hydrochemical components of global atmospheric precipitation, river water, lake water, and seawater, Gibbs concludes that the three controlling factors of global surface water hydrochemistry are atmospheric precipitation, rock weathering, and evaporation–crystallization [1,5]. The hydrochemical characteristics of river water are the comprehensive reflection of rock weathering and denudation, atmospheric subsidence, and element migration and transformation, which are often adopted to study the hydrological cycle, environmental change, and the hydrochemical process of river basin [6–9].

Qinghai-Tibet Plateau, the source region of Nujiang River, is well known as the "water tower of China" and is the location of the headwaters of several major rivers (e.g., Nujiang-Salween, Yellow River, Yangtze River, and Lancang-Mekong River) in Asia. The Nujiang River originates from China and flows into the Indian Ocean through Myanmar. More and more attention has been paid to the study of river hydrochemistry on the Qinghai-Tibet Plateau, mainly covering the river hydrochemistry characteristics, rock chemical weathering, and the response of the rock chemical weathering rate to global climate change [10–12]. The hydrogeochemical characteristics of surface water in the Qing-hai-Tibet Plateau have

been studied mainly in the Yarlung Tsangpo, Senge Zangbu, Nyan chu, and Naqu sections of Nujiang River, and Lancang River [13–16]. NOH Hyonjeong [17] analyzed the chemical weathering process of rocks in Yangtze River, Lancang River (Mekong River), and Salween River (Salween River) in the eastern Tibetan Plateau, and pointed out that the water chemical characteristics of the three rivers are different, among which the ions in the mainstream of the Nu River are mainly contributed by carbonate weathering. Zhenghua Tao [18] analyzed the chemical weathering of rocks in the Jinsha River, Lancang River, and Nujiang River basins, and found that Ca-HCO$_3$ dominates the water chemistry type of Nujiang River. The contribution of evaporative salt rock dissolution is small (2%), and carbonate rock chemical weathering contribution to the cations of the main stream of the Nujiang River is 78%. The contribution of the chemical weathering of silicate rocks to the Nujiang main stream is 15%. Zhao [19] found that the main controlling mechanism of water chemical characteristics of the Naqu River basin is rock weathering, and the water chemical characteristics of the upper reaches of Naqu River basin and the middle and lower reaches of the Naqu River basin, and the tributaries of Naqu River basin, are quite different.

Studies [20,21] show that climatic seasonality may influence the hydrochemical attributes, and hydrochemical studies focusing on seasonal variability in rivers emphasize the hydrology cyclic flow path as a significant determinant of hydrochemistry [22]. Distinct seasonal meteorology patterns, such as rainy and dry seasons, monsoons, and sun exposure, may significantly affect the intensity and direction of the water cycle and hydrochemical process [23–25].

Research on the hydrochemistry of the Tibetan Plateau basin mainly focuses on the large watershed scale and obtaining the average information of a large area [26,27]. In contrast, the hydrogeochemical characteristics and the main sources of geochemical components in the upper Nujiang River basin are rarely reported, especially in the Dingqing County section. Due to the influence of natural processes and human activities, terrestrial surface water quality is experiencing different degrees of deterioration, and water environmental issues have become increasingly prominent worldwide [28–30]. Considering the significance of the ecosystem and the protection of water resources, it is crucial to identify the unique pattern of hydrochemistry and dominant chemical weathering processes in this region. Based on the hydrochemical indexes of samples from Daqu River and Gaqu River in Dingqing County, this study combined statistical analysis, spatial analysis, trilinear diagram analysis, and ion ratio analysis to study the hydrogeological evolution process, which may be helpful for water resource exploitation and basin ecology protection.

## 2. Study Area

### 2.1. Overview

Dingqing County belongs to the plateau subfrigid zone or temperate semi-humid monsoon climate zone, with a moderate climate and four distinct seasons, thin air, and a significant temperature difference between day and night. The annual average temperature is 2.7~4.1 °C, the highest temperature is 10.1~11.9 °C, and the lowest temperature is −2.2~3.1 °C. The average annual rainfall is 634 mm, with freezing days ranging from 201 to 223 and snowfall days from 103 to 126. In this area, the total population is about 58,400, with a low population density, and the land use is mainly dominated by agriculture and animal husbandry. Almost 80% of the total precipitation is concentrated in the flood season (June–September). Although the snowfall mostly occurs outside of the flood season and total snowfall accounts for less than 20% of precipitation, the snowmelt flow contributes significantly to the streamflow in May and June [31].

The study area belongs to the Bangongcuo–Nujiang stratigraphic area. The strata's distribution is controlled by the regional large faults in the NW–SE direction and their secondary derived faults (Figure 1). Triassic Dingqing ophiolite, the later marine, continental sedimental strata, and Eocene intrusive rock are developed as the main strata in the area. Ranging from new to old are Quaternary sedimentary rocks, Eocene medium-fine-grained two-mica monzogranite, Cretaceous sandstone and limestone, Jurassic ophiolite,

limestone, dolomite and shale, Triassic ophiolite, Carboniferous slate and limestone, and Pre-carboniferous schist.

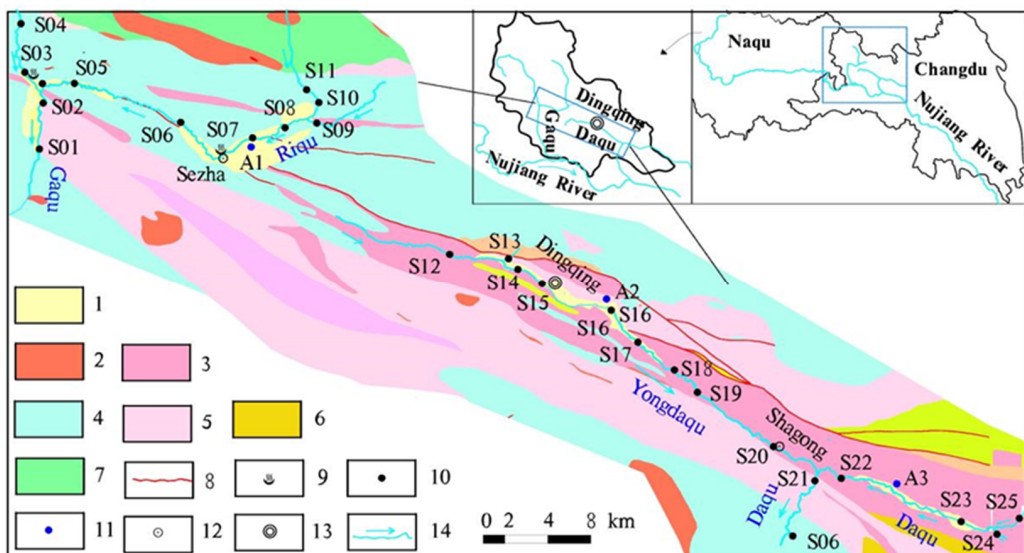

**Figure 1.** Study area and sample distribution (1. Quaternary sedimentary rocks; 2. Eocene intrusion, medium-fine-grained two-mica monzogranite; 3. Cretaceous sandstone; 4. Jurassic ophiolite, limestone, dolomite, and shale; 5. Triassic Dingqing ophiolite; 6. Carboniferous slate and limestone; 7. Pre-carboniferous schist; 8. fault; 9. hot spring; 10. water sample point; 11. atmospheric precipitation sample point; 12. town; 13. county; 14. river and flow direction).

Intrusive rocks and volcanic rocks are well developed due to the intense and frequent magmatic activity and common metamorphism and deformation of different tectonic levels, forming good metallogenic conditions. Mineral resources such as gold, silver, copper, and magnesite, and a hot spring, are distributed in this area.

*2.2. Surface Water*

Daqu and Gaqu Rivers are the two tributaries of the upper Nujiang River in the Dingqing County section. Daqu River is located in the middle of Dingqing County, with a length of about 65 km and altitude ranging from 4300 to 3790 m above sea level. As a subbranch of Daqu River, Yongdaqu River originates from the northwest and flows southward, and affluxes into Daqu River in Shagong Town. The river flows through Dingqing Town, Xiexiong Town, and Shagong Town. Gaqu River is in the north. Riqu River, as a subbranch, affluxes into Gaqu River in Chidu Town. They flow through Zhaxi Town, Sezha Town, and Chidu Town, with a length of 32 km and an altitude of 4047~3791 m above sea level. Daqu River and Gaqu River are separated into two independent basins by Qiongniela Mountain (4420 m a.s.l.). The primary recharge sources are precipitation in winter–spring and glacial meltwater and precipitation in summer–autumn.

A few hot springs developed along the strike of tensional and torsion faults. Data shows that the hot spring water's temperature ranges from 25 to 62 °C, pH 6–7.6; TDS values are from 0.4 to 1.2 g/L; and flow rates are about 4–18 L/s. The main hydrogeochemical types are $SO_4$-Ca and $HCO_3$-Ca [32].

*2.3. Groundwater*

The fissures and fractures of the bedrock and the pores and voids of Quaternary loose rocks in this area are good groundwater storage space and runoff channels. There are two groundwater types: fissure water of bedrock and pore water of loose rocks. Bedrock fissure water occurs in faults, fissures, and folds of layered or massive rock masses. Water-bearing media are mainly composed of clastic rock, carbonate rock, metamorphic sandstone, slate, marble, basalt, granite, and diorite intruded in various stages. The pore water of loose rock

mainly occurs in valley plains and branch valleys, and the water-bearing media mainly consist of alluvial and alluvial sediments, such as pebbles, drift pebbles, argillaceous pebbles, and argillaceous drift pebbles, followed by gravel, fragmentary stone, argillaceous gravel, and other diluvial sediments and fluvial deposits. The thickness of the aquifer varies from several meters to tens of meters. The occurrence condition of groundwater in the high and steep bedrock mountains is poor, and the distribution of fissure water in bedrock is sporadic and meagre. Low-lying Quaternary valleys mostly have good groundwater storage conditions. The pore water of loose rocks in valley plains and large and medium-sized branch valleys is widely distributed and abundant. Groundwater and surface water are evaporated into the atmosphere during migration and flow, which increases the moisture in the atmosphere and forms precipitation, which is then transformed into surface water and groundwater, which have a very close and frequent transformation relationship.

## 3. Materials and Methods

### 3.1. Sampling

To comprehensively consider the hydrochemical evolution of surface water in the Daqu and Gaqu basins while avoiding danger in the flood season, 25 river water samples and 3 rainwater samples were collected in July 2020. Fourteen were taken from Daqu River (including Yongdaqu River), and 11 from Gaqu River (including Riqu River). Rainwater samples were collected in Xiexiong Town and Sezha Town (see Figure 1). The river water samples were collected below 10 cm of the water surface, filtered with a 0.45 μm acetate fiber membrane within 24 h, and stored in 500 mL polyethylene sample bottles.

### 3.2. Laboratory Analysis

The sample test was completed by the Hebei Geological Testing Center. Cations such as $Na^+$, $K^+$, $Ca^{2+}$, and $Mg^{2+}$ were determined by inductively coupled plasma optical emission spectrometry (ICP-OES) Vista MPX (Varian Instruments, Walnut Creek, CA, USA). Anions such as $Cl^-$, $NO_3^-$, $SO_4^{2-}$, and $F^-$ were determined by ion chromatography (IC) Dionex™ ICS-1100 (Thermo Fisher Scientific Inc., Waltham, MA, USA). $CO_3^-$ and $CO_3^{2-}$ were determined by hydrochloric acid titration. Total Dissolved Solids (TDS) were determined by the gravimetric method. The precision of the test method is expressed as the relative standard deviation (*RSD*), which is formulated as Equation (1). The standard substance with a similar concentration to the index to be measured was selected, and the analysis was repeated 12 times to calculate the relative standard deviation between the test results of each index and the corresponding standard value. In the experiment, five groups of dual samples that covered the range of indexes to be evaluated were selected for testing, and the relative deviation (*RD*) was calculated, as shown in Equation (2). The precision and accuracy of each index are shown in Table 1. Normalized inorganic charge balance (NICB) was computed. The normalized inorganic charge balance (NICB) ranged from 1.9% to 5.2%, with an average of 3.8%. Most water samples (24 samples, 96%) had NICB values below 5%, thereby validating the accuracy of water quality determinations.

$$RSD = \frac{S}{\overline{X}} \times 100\% = \frac{\sqrt{\frac{\sum_{i=1}^{n}\left(x_i - \overline{X}\right)^2}{n-1}}}{\overline{X}} \times 100\% \tag{1}$$

$$RD = \frac{|x_1 - x_2|}{x_1 + x_2} \times 100\% \tag{2}$$

**Table 1.** Detection limit and precision of each parameter analysis method.

| Parameter | Analysis Method | Detection Limit | *RSD* [%] | *RD* [%] |
|---|---|---|---|---|
| $K^+$ | ICP-OES | 0.05 | 6.39 | 0~0.63 |
| $Mg^{2+}$ | ICP-OES | 0.003 | 4.48 | 0~1.18 |
| $Ca^{2+}$ | ICP-OES | 0.02 | 3.35 | 0~0.25 |
| $Na^+$ | ICP-OES | 0.12 | 2.69 | 0.46~3.33 |
| $Cl^-$ | IC | 0.007 | 2.16 | 0~0.94 |
| $SO_4^{2-}$ | IC | 0.018 | 2.15 | 0~0.79 |
| $NO_3^-$ | IC | 0.016 | 1.81 | 0~2.81 |
| $F^-$ | IC | 0.006 | 1.46 | 0~3.61 |
| $CO_3^{2-}$ | VOL | 5 | 3.38 | 0 |
| $HCO_3^-$ | VOL | 5 | 2.80 | 0 |

*3.3. Data Analysis*

All statistical analyses were carried out using Origin v2018 (Originlab Corparation, Northampton, MA, USA) and SPSS v17.0 (IBM Corporation, Armonk, NY, USA). A statistical program, SPSS v17.0, was used to evaluate the relationships between various physiochemical parameters. The hydrochemical program, AquaChem v10 (Waterloo hydrogeologic, waterloo, ON, Canada), and origin v2018 were used to analyze the water quality and plot the water-type graph.

**4. Results and Discussion**

*4.1. Characteristics of Hydrochemical Composition and Hydrochemical Types*

4.1.1. Major Ions

The main ions' concentration data and statistical indexes of surface water of Daqu and Gaqu Rivers are reported in Table 2. For both rivers, the order of abundance of cations was $Ca^{2+} > Mg^{2+} > Na^+ > K^+$, and $HCO_3^- > SO_4^{2-} > CO_3^{2-} > Cl^-$ for anions. $Ca^{2+}$, $Mg^{2+}$, $HCO_3^-$, and $SO_4^{2-}$ were the major component of the Total Dissolved Solids (TDS). The ranges of TDS of Daqu and Gaqu Rivers were 127.8~321.3 mg/L and 143.5~228.2 mg/L, with the mean values of 238.5 and 196.4 mg/L, respectively. These are significantly higher than the world river average value (65 mg/L) [33].

**Table 2.** Statistical results of hydrochemical parameters (in mg/L).

| River Basin | Parameters | TDS | $NO_3^-$ | $SO_4^{2-}$ | $CO_3^{2-}$ | $HCO_3^-$ | $Cl^-$ | $F^-$ | $Na^+$ | $K^+$ | $Ca^{2+}$ | $Mg^{2+}$ |
|---|---|---|---|---|---|---|---|---|---|---|---|---|
| Daqu River | Mean | 238.5 | 0.94 | 52.1 | 9.57 | 185 | 0.96 | 0.071 | 1.78 | 0.85 | 42.2 | 30.6 |
| | Median | 244.1 | 0.90 | 54.0 | 8.70 | 184 | 0.97 | 0.069 | 2.22 | 0.91 | 45.8 | 29.1 |
| | S.D. [a] | 42.4 | 0.24 | 27.7 | 2.15 | 18.8 | 0.36 | 0.029 | 1.09 | 0.19 | 12.6 | 4.64 |
| | C.V. [b] (%) | 17.78 | 25.53 | 53.17 | 22.47 | 10.16 | 37.50 | 40.85 | 61.24 | 22.35 | 29.86 | 15.16 |
| | Min | 127.8 | 0.57 | 3.28 | 5.80 | 139 | 0.35 | 0.004 | 0.06 * | 0.25 | 1.12 | 24.8 |
| | Max | 321.3 | 1.35 | 111 | 11.7 | 213 | 1.70 | 0.107 | 3.15 | 1.03 | 52.4 | 41.7 |
| Gaqu River | Mean | 196.4 | 0.56 | 73.6 | 0.79 | 113 | 0.56 | 0.055 | 1.49 | 1.37 | 37.9 | 19.3 |
| | Median | 222.1 | 0.56 | 76.7 | 0.00 | 122 | 0.54 | 0.055 | 1.32 | 1.38 | 39.8 | 21.0 |
| | S.D. [a] | 35.0 | 0.18 | 13.7 | 2.62 | 23.1 | 0.16 | 0.018 | 0.88 | 0.11 | 7.5 | 3.84 |
| | C.V. [b] (%) | 17.82 | 32.14 | 18.61 | 331.65 | 20.44 | 28.57 | 32.73 | 59.06 | 8.03 | 19.79 | 19.90 |
| | Min | 143.5 | 0.34 | 58.0 | 0.00 | 74.1 | 0.28 | 0.026 | 0.57 | 1.22 | 26.5 | 13.7 |
| | Max | 228.2 | 0.77 | 89.7 | 8.70 | 148 | 0.86 | 0.086 | 3.40 | 1.56 | 46.0 | 25.0 |

[a] S.D.: standard deviation; [b] CV: coefficient of variation. * not detected; instead, half of the detection limit.

In Daqu River, the equivalent concentration of $Ca^{2+}$ accounted for 2.0%–52.0% of the total cations, with an average of 43.2% (Figure 2). The equivalent concentration of $Mg^{2+}$ accounted for 47.1%–97.8% of the total cations, with an average of 54.8%. $Ca^{2+}$ and $Mg^{2+}$ accounted for 98.0% of the total cations. $HCO_3^-$ accounted for 54.5%–85.1% of the total anions, with an average of 68.7%, and $SO_4^{2-}$ accounted for 2.6%–39.9% of the total anions, with an average of 23.0%. $HCO_3^-$ and $SO_4^{2-}$ accounted for about 91.7% of the total anions.

For the Gaqu River, the $Ca^{2+}$ and $Mg^{2+}$ equivalent concentrations accounted for 52.6% and 44.6% of the total cations, respectively, and $HCO_3^-$ and $SO_4^{2-}$ accounted for 53.6% and 45.0% of the total anions, respectively. The proportion of $Mg^{2+}$, $Ca^{2+}$, and $HCO_3^-$ equivalent concentrations in total cations and anions of Daqu was higher than that of Gaqu, whereas the proportion of $SO_4^{2-}$ in the total anions of Gaqu was lower than that of Gaqu.

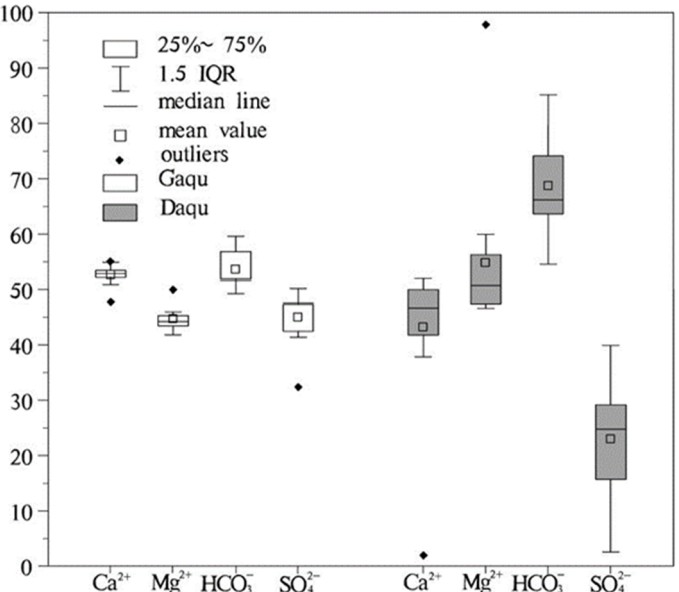

**Figure 2.** Summary statistics of main indicators' concentrations in the surface water samples studied.

Daqu River's and Gaqu River's ion coefficients of variation for $Na^+$ (61.24% and 59.06%), $SO_4^{2-}$ (53.17% and 18.61%), $F^-$ (40.85% and 32.73%), and $Cl^-$ (37.50% and 28.57%) exceeded others, which indicates that these parameters had significant spatial differences.

### 4.1.2. Spatial Variation Characteristics of Major Ions and TDS

The Yongda River flows into the Daqu River at point S20, and the Riqu River joins the Gaqu River after point S05. The mass concentrations of major ions in the surface water of Daqu and Gaqu vary steadily from upstream to downstream (Figure 3).

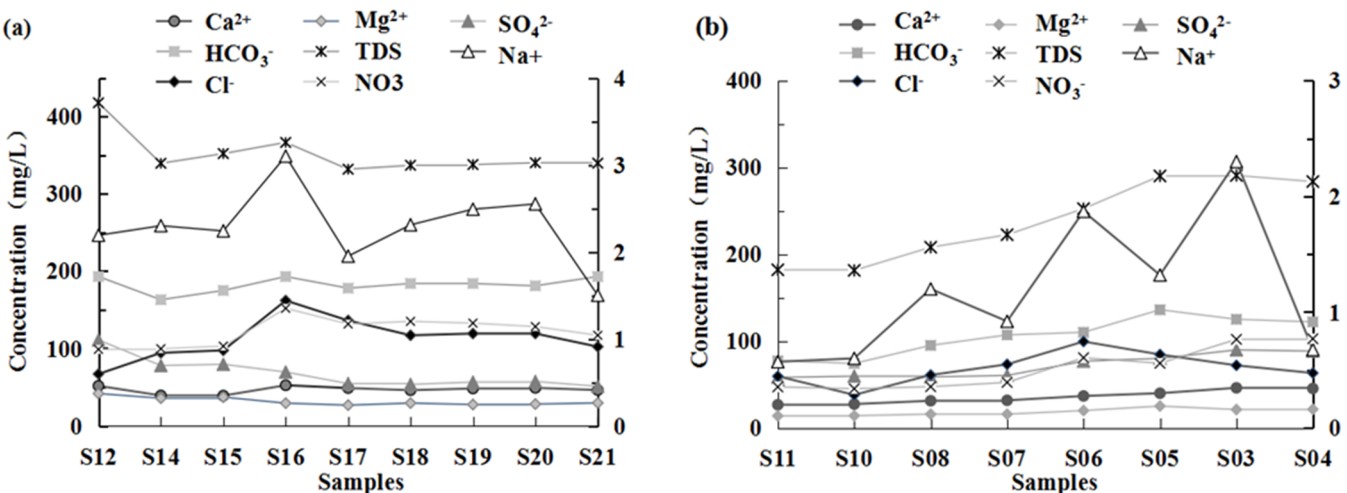

**Figure 3.** Spatial variation of major ion concentrations (**a**) for Daqu River and (**b**) for Gaqu River ($Cl^-$, $Na^+$ and $K^+$ for the right *y*-axis).

In general, for Daqu River, the content of major ions in the surface water in the flow direction slowly decreases by 24.3% (from 321.29 to 243.16 mg/L). One thing that caught

our attention is that, compared with S15, TDS has no obvious change, whereas the ion proportion has an obvious change in S16. The main increase comes from the $Ca^{2+}$ increase by 34.7% (from 38.9 to 52.4 mg/L), $Na^+$ increase by 38.4% (from 2.24 to 3.1 mg/L), and $HCO_3^-$ increase by 10.28% (from 175 to 193 mg/L). The glacier meltwater influx with low salinity along the way may be the main reason for the slow decrease in the main ion concentration, and the marked change at S16 may be caused by the mixing of hot spring water.

However, the TDS of Gaqu River increases significantly in the flow direction, from 143.46 to 222.69 mg/L, increasing by 55.2%. More than 99.7% of this growth is due to $HCO_3^-$ (from 77.1 to 122 mg/L), $SO_4^{2+}$ (from 58 to 88.3 mg/L), $Ca^{2+}$ (from 26.5 to 45.25 mg/L), and $Mg^{2+}$ (from 13.7 to 21.25 mg/L). Gaqu is probably fed by more hot spring water and less glacial meltwater than Daqu.

### 4.1.3. Hydrochemical Type

The percentage of milligrams equivalent (meq%) of the main cations and anions are plotted in the trilinear diagram (Figure 4). The composition, hydrochemical type, and evolution of cations and anions can be visually displayed or speculated upon [34–37]. Based on the triangular plots of major anions and major cations (Figure 4a,b), 14 of the 25 surface water samples can be attributed to the $HCO_3 \cdot SO_4$-$Ca \cdot Mg$ chemical type followed by $HCO_3$-$Mg \cdot Ca$ ($n = 5$), $HCO_3 \cdot SO_4$-$Mg \cdot Ca$ ($n = 3$), $SO_4 \cdot HCO_3$-$Ca \cdot Mg$ ($n = 1$), $HCO_3$-$Mg$ ($n = 1$), and $HCO_3$-$Ca \cdot Mg$ ($n = 1$) hydrochemical facies, which indicates that $Mg^{2+}$ and $Ca^{2+}$ are dominant in the Daqu River and Gaqu River. In addition, the $HCO_3^-$ ion was dominant, followed by $SO_4^{2-}$, which constituted the $HCO_3 \cdot SO_4$-$Ca \cdot Mg$ and $HCO_3$-$Mg \cdot Ca$ types for Daqu River, and the $HCO_3 \cdot SO_4$-$Ca \cdot Mg$ type for Gaqu River. Total Ionic Salinity (TIS) of water samples ranges from 4.9 to 11.6 meq/L, with an average value of 8.04 meq/L (Figure 4c). For the Gaqu river, the increase in TIS values corresponds to the increased ratios of $Mg^{2+}$ and $HCO_3^-$, and for the Daqu river, $SO_4^{2-}$ increases with the rise in TIS.

### 4.1.4. Correlation Analysis of Chemical Indexes

The correlation between components from the same source is strong; otherwise, the correlation is poor [10]. Hence, correlation analysis is often used to study the source of ions in hydrochemistry. The correlation between the two rivers is quite different (Table 3).

**Table 3.** Correlation coefficients between major ions.

|  |  | $Ca^{2+}$ | $Mg^{2+}$ | $Na^+$ | $K^+$ | $HCO_3^-$ | $SO_4^{2-}$ | $NO_3^-$ | $Cl^-$ | TDS |  |
|---|---|---|---|---|---|---|---|---|---|---|---|
|  | *Ca²⁺* | 1 | 0.550 * | 0.564 * | 0.66 ** | 0.711 ** | 0.78 ** | 0.673 ** | 0.345 | 0.891 ** |  |
|  | *Mg²⁺* | −0.20 | 1 | 0.514 * | 0.87 ** | 0.774 ** | 0.477 * | 0.44 | 0.55 * | 0.661 ** |  |
|  | *Na⁺* | 0.376 | −0.011 | 1 | 0.66 ** | 0.598 * | 0.418 | 0.382 | 0.564 * | 0.673 ** |  |
| *Daqu River* | *K⁺* | 0.61 ** | −0.367 | 0.278 | 1 | 0.849 ** | 0.514 * | 0.33 | 0.587 * | 0.771 ** | Gaqu River |
|  | *HCO₃⁻* | 0.249 | −0.193 | 0.057 | 0.307 | 1 | 0.524 * | 0.486 * | 0.561 * | 0.748 ** |  |
|  | *SO₄²⁻* | 0.385 | 0.331 | 0.376 | 0.199 | −0.271 | 1 | 0.745 ** | 0.2 | 0.745 ** |  |
|  | *NO₃⁻* | 0.385 | −0.066 | 0.464 * | 0.398 * | −0.113 | 0.429 * | 1 | 0.236 | 0.564 * |  |
|  | *Cl⁻* | 0.155 | −0.122 | 0.522 ** | 0.167 | −0.125 | 0.022 | 0.442 * | 1 | 0.455 |  |
|  | *TDS* | 0.385 | 0.376 | 0.464 * | 0.243 | −0.136 | 0.868 ** | 0.429 * | 0.066 | 1 |  |

\*\* Correlation is significant at the 0.01 level; \* Correlation is significant at the 0.05 level. The lower left triangle for *Daqu River*, and the upper right triangle for Gaqu river.

There is a strong correlation between TDS and $Ca^{2+}$, $Mg^{2+}$, $K^+$, $HCO^{3-}$, $Na^+$, $SO_4^{2-}$, and $NO_3^-$ for Daqu River samples, which indicates that $Ca^{2+}$, $Mg^{2+}$, $K^+$, $HCO_3^-$, $Na^+$, $SO_4^{2-}$, and $NO_3^-$ are the main contributors to the change in TDS. There is also a significant correlation between $Cl^-$ and $Na^+$ in the Daqu River, with a coefficient of 0.522, which indicates that it has a common main source, possibly from the weathering and dissolution of halite.

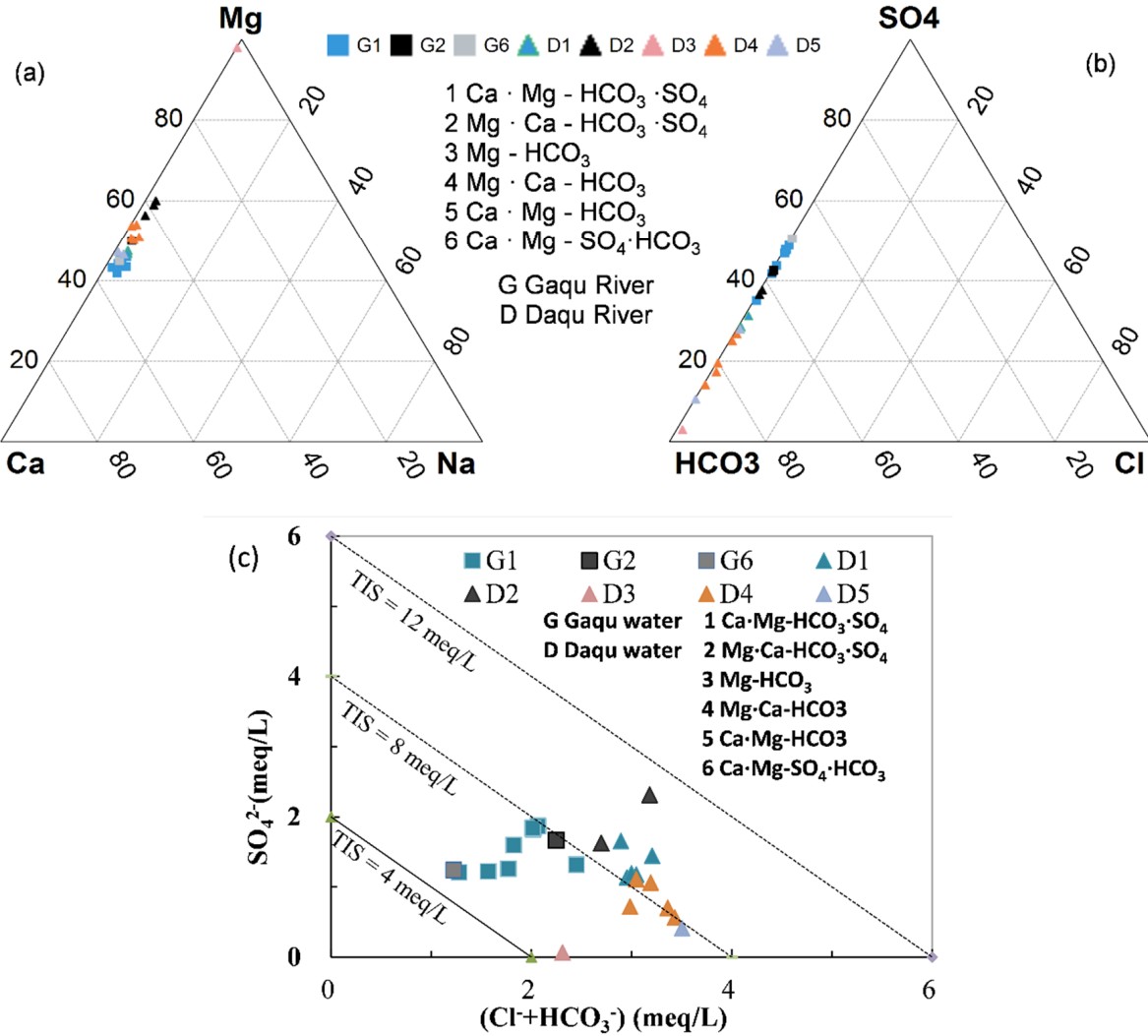

**Figure 4.** Cation and anion trilinear diagram (**a**,**b**) and TIS salinity diagram (**c**) of Daqu and Gaqu Rivers' water samples.

For Gaqu River, TDS is significantly correlated with $Ca^{2+}$, $Mg^{2+}$, $K^+$, $HCO_3^-$, and $SO_4^{2-}$, and the coefficients are 0.891, 0.661, 0.673, 0.771, 0.748, and 0.745, respectively, which indicates that the change in these ions mainly controls the shift in TDS. The significant correlation of $HCO_3^-$, $Ca^{2+}$, and $Mg^{2+}$ (0.711 for $HCO_3^-$-$Ca^{2+}$, 0.774 for $HCO_3^-$-$Mg^{2+}$, and 0.550 for $Ca^{2+}$-$Mg^{2+}$) indicates that the three ions have a common source, which mainly comes from the weathering and dissolution of dolomite. $SO_4^{2-}$ has a significant correlation with $Ca^{2+}$ (0.78), indicating that it mainly comes from the weathering dissolution of gypsum.

### 4.2. Analysis of Main Ion Controlling Factors
#### 4.2.1. Water–Rock Interaction

Water flows upstream to downstream, constantly exchanging material and energy with the surrounding environment. The Gibbs diagram often represents the hydrogeochemical processes of natural water bodies, and the main ion sources and controlling effects in water bodies can be inferred [38–40]. The Gibbs diagram is intuitively divided into three elements: evaporation–crystallization, rock weathering, and meteoric precipitation. The lower right corner of the figure is the control area of atmospheric precipitation, with low TDS concentration and a ratio of $Cl^-/(Cl^- + HCO_3^-)$ and $Na^+/(Na^+ + Ca^{2+})$ that is close to 1. In the middle of the left-hand side is the weathering zone, where TDS values are generally between 70 and 300 mg/L, and the ratios of $Cl^-/(Cl^- + HCO_3^-)$ and

$Na^+/(Na^+ + Ca^{2+})$ are generally less than 0.5. The upper right corner is the evaporation–concentration controlled area, in which the TDS concentration is generally higher than 300 mg/L, and the ratios of $Cl^-/(Cl^- + HCO_3^-)$ and $Na^+/(Na^+ + Ca^{2+})$ are between 0.5 and 1.

Water sample test data for Daqu and Gaqu Rivers were plotted into Gibbs diagrams (Figure 5). Water samples in this area have moderate TDS values and quite low ratios of $Cl^-/(Cl^- + HCO_3^-)$ (mean 0.01) and $Na^+/(Na^+ + Ca^{2+})$ (mean 0.03). The water samples all fall into the rock weathering control area, which indicates that the main ions of surface water in Gaqu and Daqu Rivers are mainly controlled by water–rock action and hardly controlled by atmospheric precipitation. This result may be explained by the fissure bedrock and pore loose rock groundwater system. Rainfall and snow/ice melting water infiltrates as the groundwater recharges, and sufficient water–rock reaction occurs before discharge. It also seems possible that these results may be due to the sampling time. The sampling activities took place in July 2020, and there was no noticeable rainfall and storm event before this time. At that moment, groundwater discharge was the dominant constituent of the river water.

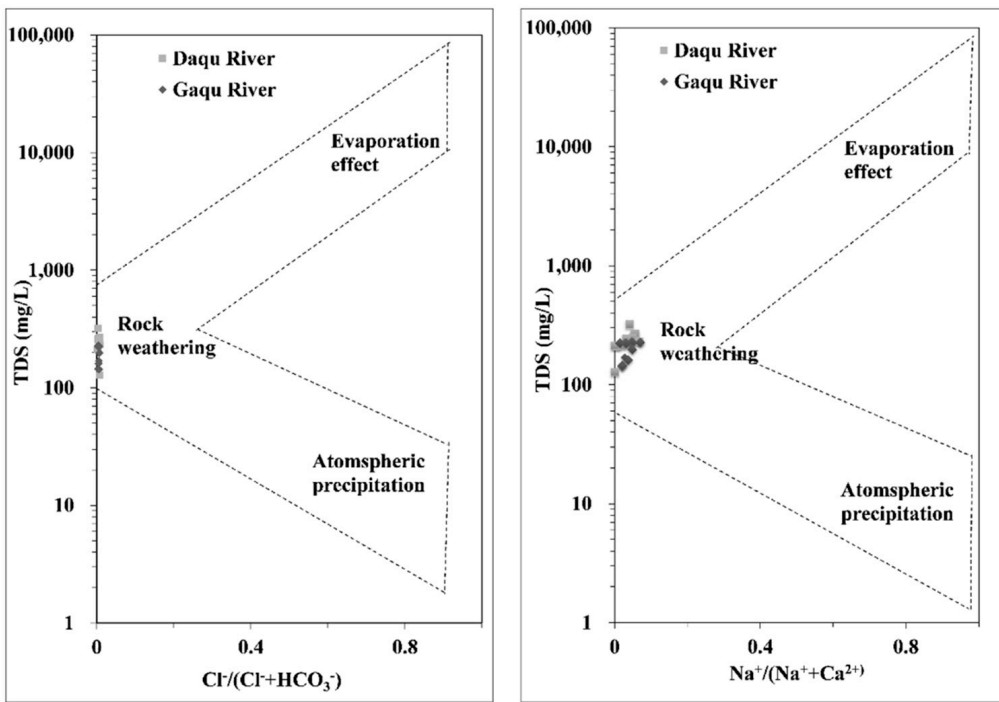

**Figure 5.** Gibbs plots of the Daqu and Gaqu River water samples.

Different ion compositions indicate various types of rock weathering, so the primary ion equivalent ratio in the water body is commonly used to study the main ion sources in the water body [41–43]. Meybeck [44] showed that the contribution of carbonate rocks to global river TDS is about 50%, and that the weathering of evaporated halite and silicate rock contributes about 17.2% and 11.6%, respectively. For instance, $Ca^{2+}$ and $Mg^{2+}$ mainly originate from the weathering of carbonates, silicates, and evaporites, Na and K from the weathering of evaporites and silicates, $HCO_3^-$ from carbonates and silicates, and $SO_4^{2-}$ and $Cl^-$ from evaporites [45]. Combinations of the three representative lithologies, i.e., evaporites, carbonates, and silicates, can be displayed on plots of Na-normalized molar ratios (Figure 6). The mixing diagram usually shows the origins of the ions produced by chemical weathering in a basin with complex lithology [46,47].

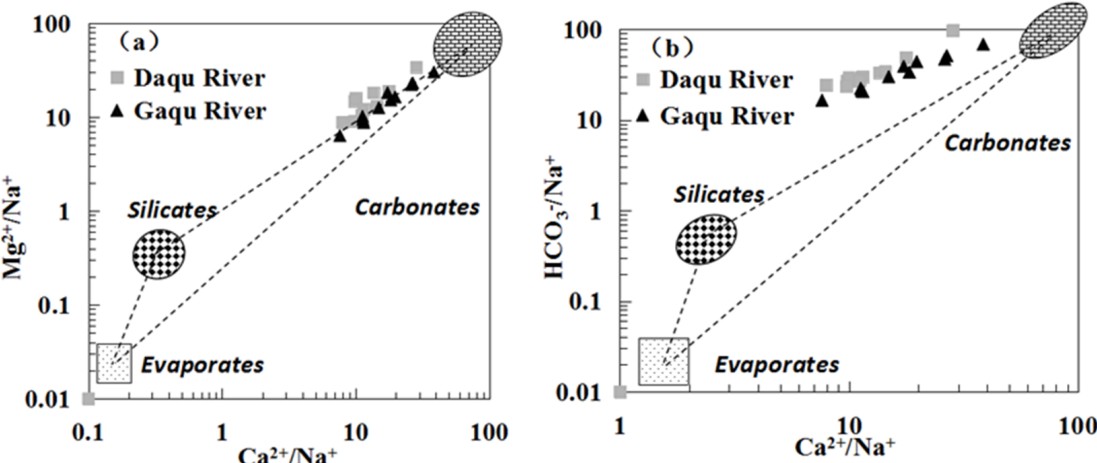

**Figure 6.** Molar ratio bivariate plots of (**a**) Na-normalized Ca and HCO₃ and (**b**) Na-normalized Ca and Mg. End-member compositions of carbonate, silicate, and evaporite are from Gaillardet J [46].

Samples from both Daqu River and Gaqu River are plotted between the carbonates and silicates end-members. This shows that the surface water in the study area is mainly controlled by the weathering of carbonate rocks and silicate rocks, which is consistent with the regularity with which the lithology distribution in the study area is dominated by carbonate rocks and the sporadic distribution of silicate. It is worth noting that the selected end-member data are not from the study area, so the sample did not fall strictly in the triangle area, but the qualitative result is still of reference significance.

### 4.2.2. Cation Exchange

The plot of $(HCO_3^- + SO_4^{2+} - Ca^{2+} - Mg^{2+})$ (meq) and $(Na^+ - Cl^-)$ (meq) shows the cation exchange reactions between Ca, Mg, and Na [47,48]. The ratio relation can reflect the control effect of cation exchange. If cation exchange is an important process controlling the ionic composition of the samples, the relationship between these parameters should be linear with a slope of 1:1. The Chlor-alkali index can determine the direction and strength of cation exchange in groundwater. As shown in Equations (3) and (4), if CAI1 and CAI2 are both positive, it indicates that $Na^+$ and $K^+$ in groundwater exchange with $Ca^{2+}$ and $Mg^{2+}$ in the surrounding rock, and if they are negative at the same time, exchange in the reverse direction occurs [49].

$$CAI1 = \frac{Cl^- - (Na^+ + K^+)}{Cl^-} \tag{3}$$

$$CAI1 = \frac{Cl^- - (Na^+ + K^+)}{Cl^-} \tag{4}$$

As shown in Figure 7a,b, all the water samples of Gaqu lie on the $-1:1$ line, and CAI1 and CAI2 are both small negative numbers, which shows that weak reverse cation exchange occurred in the two rivers. The dissolved $Ca^{2+}$ and $Mg^{2+}$ in the carbonate rock in the surface water exchanged with the $Na^+$ and $K^+$ in the surrounding rock of the tunnel. The adsorption of $Ca^{2+}$ and the release of $Na^+$ explain the increase in $Na^+$ in the Gaqu water sample, which also leads to changes in the chemical composition of the water. The lithology of Gaqu mainly includes ophiolite, limestone, dolomite, and shale. There are silicate minerals and clay minerals, such as feldspar and quartz, in these rock formations. According to the geological survey, hot springs provide $SO_4^{2-}$, $Na^+$, and $K^+$. According to the size of the Chlor-alkali index, the intensity of cation adsorption alternate action in the surface water of Gaqu is greater than that of Daqu.

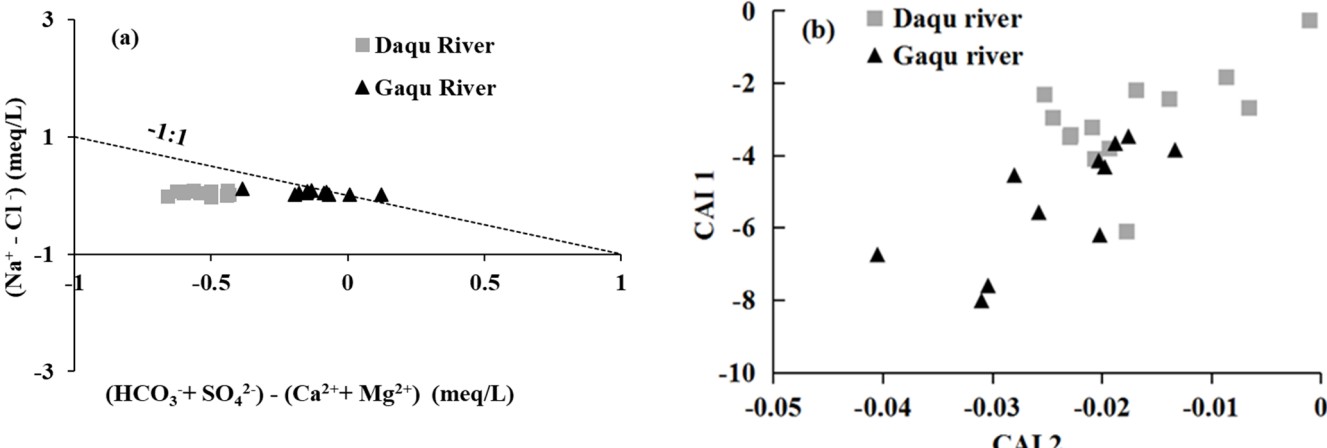

**Figure 7.** Analysis of ion exchange (**a**) for the plot of $(HCO_3 + SO_4) - (Ca + Mg)$ vs. $Na - Cl$, and (**b**) for the Chlor-alkali index plot.

### 4.2.3. Anthropogenic Activity Influence

Industrial production and domestic and agricultural activities will release $NO_3^-$ and $Cl^-$ ions, which affect the hydrochemical evolution [49]. Therefore, the water affected by anthropogenic activities presents high $Cl^-/Na^+$ and $NO_3^-/Na^+$ ratios [50,51]. The ratios of $Cl^-/Na^+$ and $NO_3^-/Na^+$ in the two basins were relatively low at most points (Figure 8), indicating that the surface water in the two basins is weakly affected by human activities, which mainly reflected the hydrochemical evolution under natural conditions. Only sample point S24 of Daqu had higher ratios of $Cl^-/Na^+$ and $NO_3^-/Na^+$ (0.96 and 0.88, respectively). Zizhu Temple, a famous temple in eastern Tibet, is located upstream of this point; a greater number of visits could result in abnormal water quality.

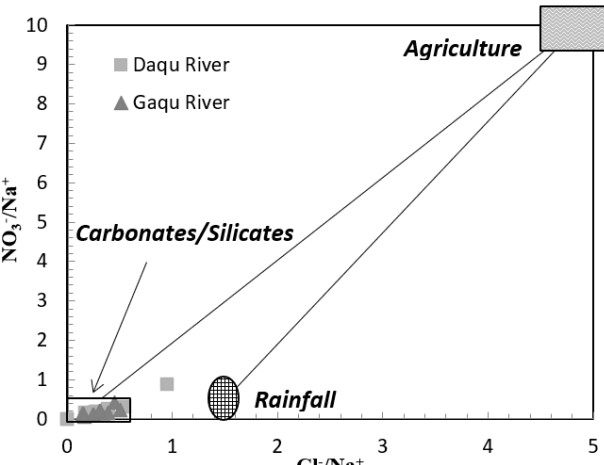

**Figure 8.** $NO_3/Na$ vs. $Cl/Na$ mixing diagram between rainfall, carbonate/silicate weathering, and agriculture end-members.

### 4.3. Main Ion Source Analysis

The ion ratio can be used to deduce the leaching effect in the study area and then to study the main ion sources in the water. The $(Ca^{2+} + Mg^{2+})/HCO_3^-$ equivalent ratio and the ratio of $Ca^{2+}/Mg^{2+}$ in water are often used to study the source of $Ca^{2+}$ and $Mg^{2+}$ [52]. If $Ca^{2+}$ and $Mg^{2+}$ are only derived from dissolved weathering of aquifer carbonate rocks, the $(Ca^{2+} + Mg^{2+})/HCO_3^-$ equivalent ratio should be 0.5 [53]. The $(Ca^{2+} + Mg^{2+})/HCO_3^-$ average equivalence ratio of Daqu and Gaqu ranged from 1.2 to 2.1, with an average value of 1.7 (Figure 9a), and the $Ca^{2+}/HCO_3^-$ ranged from 0.6 to 1.2, which means $Ca^{2+}$ and $Mg^{2+}$ require more $SO_4^{2-}$ and/or $Cl^-$ to achieve balance. The sample plotted close to the

1:2 line, which means the mole numbers of $Ca^{2+}$ and $Mg^{2+}$ are similar. The result indicates that, in addition to carbonates weathering, other sources of $Ca^{2+}$ and $Mg^{2+}$ also exist in the study area.

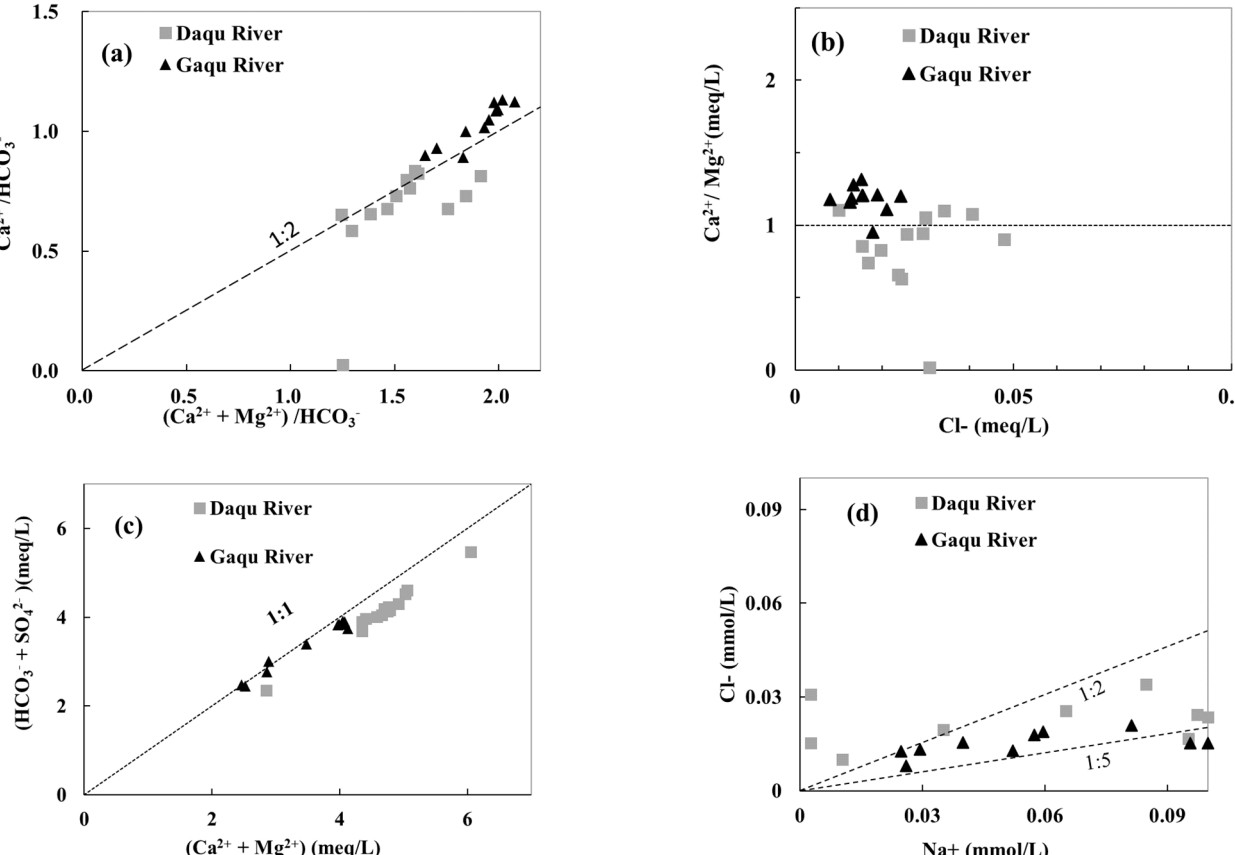

**Figure 9.** Bivariate plots of the major ions in groundwater: (**a**) $(Ca^{2+} + Mg^{2+})/HCO_3$ versus $Ca^{2+}/HCO3$; (**b**) $Ca^{2+}/Mg^{2+}$ versus $Cl^-$; (**c**) $HCO_3^- + SO_4^{2-}$ vs. $Ca^{2+} + Mg^{2+}$; (**d**) $Cl^-$ versus $Na^+$.

In general, $Ca^{2+}/Mg^{2+} = 1$ indicates that $Ca^{2+}$ and $Mg^{2+}$ are derived from the dissolution of dolomite $(Ca\,Mg\,(CO_3)_2)$, and $1 < Ca^{2+}/Mg^{2+} < 2$ indicates the occurrence of the dissolution of calcite $(CaCO_3)$ [42], whereas $Ca^{2+}/Mg^{2+} < 1$ indicates the occurrence of the dissolution of magnesite $(MgCO_3)$. Except for S05, $Ca^{2+}/Mg^{2+}$ ratios of Gaqu water samples are all between 1 and 1.5 (Figure 9b), which indicates that calcite dissolution occurs in Gaqu. $Ca^{2+}/Mg^{2+}$ ratios of Daqu water samples are distributed around 1. Most are lower than 1, which, combined with the geological background of Daqu, may be related to weathering and dissolution of magnesite in the basin.

The equivalent concentration ratio of $(Ca^{2+} + Mg^{2+})$ to $(HCO_3^- + SO_4^{2-})$ is usually used to judge the control of the dissolution of calcite, dolomite, or gypsum on the hydrochemistry of the basin [54,55]. The equivalent concentration ratio in water samples will be 1:1 if the chemical composition of the water is controlled by the dissolution of calcite, dolomite, and/or gypsum. The equivalent concentration ratios of surface water samples in Gaqu are all around the 1:1 line (Figure 9c), indicating that the hydrochemical composition of the basin is controlled by the dissolution of calcite, dolomite, and/or gypsum. The ratios in Daqu samples are all below 1:1, indicating that the equivalent concentration of $Ca^{2+} + Mg^{2+}$ is higher than that of $HCO_3^- + SO_4^{2-}$, which means that a part of $Ca^{2+}$ and $Mg^{2+}$ in Daqu samples should be derived from the dissolution of silicate rock (e.g., albite).

In most water samples in Gaqu and Daqu basins, the molar concentration ratio of $Cl^-/Na^+$ is less than 1 (Figure 9d), indicating that $Na^+$ concentration was higher than $Cl^-$ concentration. Except for S13, S24, and S25 in Daqu (0.08, 1.05, and 0.17, respectively), the $Na^+/Cl^-$ molar ratios in all samples range from 1.8 to 6.58. Combined with the actual

situation of the research area, in which there is no evidence of halite, $Na^+$ is mainly derived from the weathering of silicate rocks or hot spring water.

Therefore, despite carbonate (calcite, dolomite) being the main factor, dissolution of silicate and magnesite also occur in the study area, and the processes are as follows:

$$CaMg(CO_3)_2 \text{ (dolomite)} + 2CO_2 + 2H_2O \rightarrow Ca^{2+} + Mg^{2+} + 4HCO_3^-$$

$$CaCO_3 \text{ (calcite)} + CO_2 + H_2O \rightarrow Ca^{2+} + 2HCO_3^-$$

$$2NaAlSi_3O_8 \text{ (albite)} + 2CO_2 + 11H_2O \rightarrow Al_2Si_2O_5(OH)_4 + 2Na^+ + 4H_4SiO_4 + 2HCO_3^-$$

$$MgCO_3 \text{ (magnesite)} + CO_2 + H_2O \rightarrow Mg^{2+} + 2HCO_3^-$$

## 5. Conclusions

This study tested the main ions of 25 river water samples of Daqu and Gaqu Rivers, and the hydrochemical composition and main ion controlling factors were analyzed. The main conclusions can be drawn as follows:

1.  Daqu and Gaqu, as two tributaries of the upper reaches of Nujiang River, are mainly composed of $Ca^{2+}$ and $Mg^{2+}$ for cations (accounting for about 98%) and $HCO_3^-$ and $SO_4^{2-}$ for anions (accounting for about 91.7%). The main hydrochemical types are $HCO_3 \cdot SO_4$-$Ca \cdot Mg$ and $HCO_3 \cdot SO_4$-$Mg \cdot Ca$.
2.  The influence of atmospheric precipitation and anthropogenic activities is weak.
3.  The carbonated water–rock reaction is the main $Ca^{2+}$, $Mg^{2+}$, and $HCO_3^-$ source, and hot springs act as the main source of $SO_4^{2-}$ and supplement of $Ca^{2+}$, $Mg^{2+}$, and $HCO_3^-$.
4.  Weak reverse cation exchange occurs in both rivers and is more intensive in the Gaqu River.
5.  $Mg^{2+}$ from magnesite dissolution exists in the Daqu River basin.
6.  Daqu River receives more low-mineralized glacier meltwater along the flow, whereas Gaqu River receives more high-mineralized hot spring water.

**Author Contributions:** Conceptualization, M.W. and L.Y.; methodology, M.W.; software, M.W.; validation, L.Y.; formal analysis, Q.L.; investigation, J.L.; resources, Q.L.; data curation, J.L.; writing—original draft preparation, M.W.; writing—review and editing, L.Y.; visualization, M.W.; supervision, Q.L.; project administration, M.W. All authors have read and agreed to the published version of the manuscript.

**Funding:** This research was funded by the Geological survey project (DD20190534 and DD20221754) of the China Geological Survey.

**Institutional Review Board Statement:** Not applicable.

**Informed Consent Statement:** Not applicable.

**Data Availability Statement:** The study did not report any data.

**Conflicts of Interest:** The authors declare no conflict of interest.

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
