# Peer review of "Hydrochemical Characteristics and Controlling Factors of Surface Water in Upper Nujiang River, Qinghai-Tibet Plateau"

_minerals, doi:10.3390/min12040490_

Round 1

Reviewer 1 Report

The purpose of the manuscript “minerals_1640423” is to investigate the geochemistry of Daqu River and Gaqu River in Dingqing county, two tributaries of the upper Nujiang river,  through the use of spatial analysis, trilinear diagram analysis, and ion ratio analysis.

The paper appears well-structured; however, some sections must be improved.  Therefore, I believe the manuscript should be published only after major revision.

Comments

The introduction is too short and does not take into account many general factors responsible for mineralization and water pollution. Read and add the following books and work in the references:

Appelo, C.A.J. and Postma, D., 2004. Geochemistry, groundwater and pollution. CRC press.

Langmuir, D., 1997. Aqueous environmental geochemistry (No. 551.48 L3.).

Fuoco, I., De Rosa, R., Barca, D., Figoli, A., Gabriele, B. and Apollaro, C., 2022. Arsenic polluted waters: Application of geochemical modelling as a tool to understand the release and fate of the pollutant in crystalline aquifers. Journal of Environmental Management, 301, p.113796.

in figure 1 the location of the rainwater is missing

in table 1 there are analyzes in ICP while in the text we talk about flame atomic absorption spectrometry. Correct either the table or the text.

To evaluate the chemical composition of the water it is not enough to use the Piper diagram because it does not take into account (as proposed by the authors) salinity, I suggest using a TIS salinity diagram, as proposed by:

Apollaro, C., Di Curzio, D., Fuoco, I., Buccianti, A., Dinelli, E., Vespasiano, G., ... & De Rosa, R. (2022). A multivariate non-parametric approach for estimating probability of exceeding the local natural background level of arsenic in the aquifers of Calabria region (Southern Italy). Science of The Total Environment, 806, 150345.

The proposed ion exchange diagram is not very effective. It is recommended to use a diagram with CAI data. See for example the work:

Peng, Chen, Yuanming Liu, Huiyu Chen, Qiaowei Yuan, Qingzhi Chen, Shilong Mei, and Zhonghu Wu. "Analysis of Hydrogeochemical Characteristics of Tunnel Groundwater Based on Multivariate Statistical Technology." Geofluids 2021 (2021).

Discussions and conclusions need to be rewritten taking into account previous comments

Recommended works must be added in the bibliography

Reviewer 2 Report

Summary of study

This paper investigated the hydrochemical characteristics of two rivers in China. They reported the concentration of various cations and ions including Ca2+, Mg2+, HCO3-, and SO42- from the river. They also investigated the effect of precipitation, hot springs, and carbonated rock reaction on the concentration of ions at different positions in the river. 

Major comments

1. I suggest the authors also include some related studies or article reviews with a focus on similar research areas in the introduction section. The introduction is very short and did not cover any related studies that reported the hydrochemical characteristics of rivers. Some examples of related studies are:

  1. Hua, K., Xiao, J., Li, S., & Li, Z. (2020). Analysis of hydrochemical characteristics and their controlling factors in the Fen River of China. Sustainable Cities and Society52, 101827.
  2. Zong-Jie, L., Ling-Ling, S., Juan, G., & Zong-Xing, L. (2022). Hydrochemical patterns indicating hydrological processes with the background of changing climatic and environmental conditions in China: a review. Environmental Science and Pollution Research, 1-16.

2. The samples were not collected seasonally. Although the temperature seems not to have a dramatic change during the year, a brief discussion of the effect of various environmental conditions, such as humidity and sun exposure needs to be discussed in the manuscript. 

3. In the method section, the formula and approach that precision (reported in Table 1) was calculated needs to be added. 

4. In the method section, the approach for the calculation of TDS needs to be added. 

5. Have authors investigated the ratio of Ca2+ and HCO3- besides the Ca2+ + Mg2 + versus HCO3- that is plotted in Figure 9a? 

6. Since at a high concentration of ions the chance of complexation and precipitation will increase, have the authors also collected any sample from the deeper part of the river to check the variability at different depths rather than different positions along the river? If so, please add and discuss the data. 

Minor comments

  1. Line 205 and 206, notation needs to be corrected. 
  2. Line 320, since the Figure is plotted for the Cl- versus Na+ concentration, adding the ratio of Cl-/Na+ can be useful for the reader. 

Round 2

Reviewer 1 Report

Remarks from both reviewer have been correctly addressed, and the paper is now more focuse on his core topic
In my opinion it is now acceptable.
Best regards

Author Response

Thanks for your constructive comments.

Reviewer 2 Report

This paper investigated the hydrochemical characteristics of two rivers in China. They reported the concentration of various cations and ions including Ca2+, Mg2+, HCO3-, and SO42- from the river. They also investigated the effect of precipitation, hot springs, and carbonated rock reaction on the concentration of ions at different positions in the river. 

Authors have significantly improved the manuscript:

  1. Introduction was majorly revised and included some related studies with a focus on similar research areas. They extended the length of introduction to four paragraphs and the suggested articles were added to the manuscript.
    1. Hua, K., Xiao, J., Li, S., & Li, Z. (2020). Analysis of hydrochemical characteristics and their controlling factors in the Fen River of China. Sustainable Cities and Society52, 101827.
    2. Zong-Jie, L., Ling-Ling, S., Juan, G., & Zong-Xing, L. (2022). Hydrochemical patterns indicating hydrological processes with the background of changing climatic and environmental conditions in China: a review. Environmental Science and Pollution Research, 1-16.

  1. References were added in the introduction to discuss the seasonal variations. They also added explanation about their approach and limitations for sample collection.

  1. In the method section, they added the formula for precision and also the TDS measurements. I would still recommend to add more explanation that if the similar samples were measured multiple times and averaged with different sample measurements for accuracy measurements? Or each sample was prepared independently and then the measured once for calculation of the precision.  
  2. Revised Figure 9a is more informative regarding the ratio of Ca2+ and HCO3- besides the Ca2+ + Mg2 + versus HCO3-.
